# Social relations and mental ill-health among newly arrived refugees in Sweden: A cross-sectional study

Elisabeth Mangrio[1,2]*, Mathias Grahn[3], Slobodan Zdravkovic[1,2], Carin Cuadra[4]

**1** Faculty of Health and Society, Department of Care Science, Malmö University, Malmö, Sweden, **2** Malmö Institute for Studies of Migration, Diversity and Welfare [MIM], Malmö University, Malmö, Sweden, **3** Unit for Safety and Security, Municipality of Malmö, Malmö, Sweden, **4** Department of Social Work, Malmö University, Malmö Sweden

\* elisabeth.mangrio@mau.se

**Data Availability Statement:** Due to research ethical requirements, we are prohibited from anonymizing the data and thus we need to follow the GDPR in processing it. Passing the research

## Abstract

### Background

Previous research indicates that social relations have an impact on the well-being of refugees and that well-being is important for effective integration into the host country. Few studies in Sweden have, to the best of our knowledge, looked at the association between social relations and mental ill-health among newly arrived refugees. The aim is to investigate what effect social relations have on the mental health of newly arrived refugees in the south of Sweden.

### Methods

A cross-sectional study was conducted in Scania, the southernmost county of Sweden, between February 2015 and February 2016. The study population consisted of newly arrived adult refugees speaking Dari or Arabic, who received the civic and health information that is part of an introduction course for all newly arrived refugees.

### Results

Individuals who rarely met with friends had higher odds of experiencing mental ill-health [OR = 1.70, 95% CI, 1.03–2.82] than individuals who frequently spent time with friends. Furthermore, individuals who seldom attended social/community meetings or activities in an organisation or group, such as a sports association or another kind of association, a church, a mosque, or women's or men's meetings, had higher odds of mental ill-health (OR = 1.58, 1.1–2.28), compared to those who frequently did so.

### Conclusions

The study suggests a link between spending time with friends, as well as engaging in social/community activities, and the mental health of newly arrived refugees in the southernmost county of Sweden, which is one of the counties in Sweden that received the highest number of refugees. The results are in line with a previous study on the same subject in Sweden.

data on to a third party would be a violation of articles 5.1b (purpose limitation) and 9.1 (since we have no applicable exception in article 9.2) in the GDPR. Contacts to the ethical board of Sweden can be taken through the following email and phone-number: registrator@etikprovning.se and +46-104750800.

**Funding:** This research was supported by European Refugee Fund (The main applicant was Carin Cuadra and co-applicant was Slobodan Zdravkovic). The funders had no role in study design, data collection and analysis, decision to publish, or preparation of the manuscript.

**Competing interests:** The authors declare that no competing interests exist.

## Introduction

Social relations have been shown to be of great importance for the health and well-being of refugees regardless of culture, age, and host country [1]. A study from Australia showed that being part of informal social networks protected African refugees from mental stress [2]. The refugee families that had family and friends close by, felt better both **socially** and emotionally [2]. Another study from Canada showed that refugees that had close contact with other people from their own country of birth after flight, were more protected against stress and had a decreased risk of leaving the current host country for another one [3]. According to yet another study, refugees keeping contact with family and friends from their country of birth were doing better during their resettlement process in the host country [4]. Furthermore, an American study showed that the well-being of refugee women had improved during their resettlement, because of the impact that faith and religion played in their life [1]. Since a lot of the social activities and meetings were connected to the church, the church was an important place for social gatherings that could positively affect the integration into the society [1]. Another study confirms the importance of social gatherings within religious organisations as a resource after arrival in the host country and claims that this resource could be seen as an integral part of the integration work in the host country [5].

Refugees meet different kinds of challenges during their establishment in the new society, and feeling socially excluded from society in the host country is one of those challenges [6]. In addition, language difficulties are a barrier for both adults and children, keeping them from being socially engaged with people in the host country [5]. Refugees in Sweden have to adjust to another kind of social life and some families struggle with being separated from their families in different ways, something that affects their mental well-being [7]. Recent research in Sweden has also shown that crowded living is common among newly arrived refugees [NAR] in Scania, the southernmost county in Sweden, and that this has an association with risk for mental ill-health [8].

Research indicates that social relations have an impact on the well-being of refugees and that well-being is important for successful integration into the new host country. For example, recent research in Sweden, conducted at an earlier point in time than the present study and focusing on Iraqi refugees, affirms the impact that social relations have on mental well-being among refugees [9]. However, to the best of our knowledge, no earlier studies in this field have been conducted among all NAR within the county of Scania, which is one of the counties in the country that receive the highest number of NAR. It is, therefore, of great importance to determine how social relations after flight affects mental health among NAR in the southernmost county of Sweden. The aim of this study is to investigate whether there is any association between social relations after arrival and mental health among NAR in Scania.

## Material and methods

### Selection process

The study population consisted of all NAR in Scania, the southernmost county of Sweden, who received civic and health information during the time period starting February 2015 and finishing February 2016 [10]. Civic and health information is given in the migrants' mother tongue, through a comprehensive introduction programme offered to all migrants as soon as possible upon arrival in the region. The programme is comprised of mandatory civic orientation supplemented with health communication. The programme also organises and facilitates visits to different arenas in the local community through the project "Welcome to Skåne" [11]. During the period covered by the present study, there were mainly Arabic- and Dari-speaking NAR within this programme.

## Data collection

The questionnaires were distributed at the same time to all NAR that received civic and health information as part of the mandatory public integration support programme in the southern-most county of Sweden. Around 1,700 questionnaires were translated and handed out to refugees speaking Dari or Arabic, and 681 questionnaires were returned [drop-out rate 60.5%, response rate 39.5%]. Before the NAR participated in the survey, they had to sign an informed written consent. The questions were self-administered, and paper-based, covering aspects of health, educational level, sleep habits, mental health, social relationships, housing, employment, and health care access.

**Dependent variable.**   The General Health Questionnaire (GHQ-12) was used to assess the risk of mental ill-health. It consists of 12 questions that measure the well-being of a person, including depressive symptoms, sleep habits, anxiety, and cognitive functioning. Each item is a statement that is scored on a scale that goes from "not at all" to "much more than usual". The questionnaire contains both positive and negative questions. Positive questions (items 1, 3, 4, 7, and 12) are scored inversely. Conventionally, the scale is made negative, so that a high value means low psychological well-being and a low value high psychological well-being. We used a 0,0,1,1 scoring model, and Goldberg's original recommendation of a score sum of $\geq 3$ was used. The respondent's overall GHQ-12 score was categorised as poor if three or more items indicated poor psychological health.

The scale is used to indicate risk for the development of mental illness and is extensively used and suited for larger epidemiological studies [12]. It has been tested for psychometric properties and could be considered a reliable tool for the assessment of risk for mental ill-health [13].

**Independent variables.**   Social relations were assessed by means of three different questions:

The first question asked about how often the person was together with close family, such as parents, sisters or brothers, or children over 18 years old. The second question asked about how often the person met and spent time with friends or relatives. The third question asked about how often in their spare time the person attended meetings or activities in an organisation or group, such as a sports association or another kind of association, a church, a mosque, or women's or men's meetings. For all three questions, the answer options were: "daily", "every week but not daily", "every month but not every week", and "more seldom or never". And for each question, the first three answer options were grouped into "meeting close family often", "meeting friends and relatives often", and "attending meetings and activities often", respectively, and the last one into "meeting close family seldom", "meeting friends and relatives seldom", and "attending meetings and activities seldom", respectively.

In order to rank the level of social relations and how this is associated with mental health, the three questions described above were used to derive a combined measure of social relations. The variables were grouped into four categories as follows:

1. Answering "seldom" for all the above stated variables, that is, seldom being together with close family, such as parents, sisters or brothers, or children over 18 years old; seldom meeting and spending time with friends or relatives; and seldom attending meetings or activities in an organisation or different groups.

2. Answering "seldom" for two out of three of the above stated variables.

3. Answering "seldom" for one out of three of the above stated variables.

4. Answering "often" for all three of the above stated variables. This category served as the reference category in the analysis.

The question on educational level was categorised into low level (≤ 9 years), medium level (10–12 years), and high level (>12 years).

Age was categorised into the following five groups: 18–34 years, 35–44 years, 45–54 years, 55–64 years, and 65–80 years. Sex was self-reported as either female or male.

In order to assess crowded living, the following two questions were asked: How many rooms do you have? How many people live in your home? The housing was considered crowded if more than two people were living in the same bedroom.

Housing conditions were assessed to be good if the person (a) lived in a rental flat without a creditor or (b) owned a house or a flat. The housing conditions were seen as unstable if the person (a) lived in a rented flat with a creditor, (b) lived in a sub-let flat, (c) lived with family or friends, or (d) lived in accommodation facilitated by the government.

## Analysis

The descriptive statistics are presented in terms of frequencies and percentages. In order to analyse the association between social relations and mental health, a logistic regression model was used to calculate odds ratios and 95% confidence intervals. The influence of confounders, such as educational level, age, gender, crowded living, and housing conditions, was handled by using the multiple logistic regression model in order to adjust the crude odds ratios. Statistical analyses were conducted by SPSS version 22.

## Ethical considerations

Ethical approval was received from the Regional Ethical Committee in Lund, Sweden (reg.no. 2014/285), before the study was conducted. All informants had to sign a written informed consent prior to participation in the study. As the study involved human participants, it was performed in accordance with the Declaration of Helsinki.

## Results

In total, 681 NAR answered and returned the questionnaire. Around 50.3% had a high educational level, 23.7% a medium level, and 26.0% a low educational level. With regard to social relations, 37.2% were seldom together with close family such as parents, sisters or brothers, or children over 18 years old; 14.7% seldom met and spent time with friends and relatives; and 46% seldom attended meetings or activities in an organisation or group, such as a sports association or another kind of association, a church, a mosque, or women's or men's meetings. Frequencies and percentages for all studied variables are presented in Tables 1 and 2. It was not possible to conduct any proper drop-out analysis since all respondents were anonymous. However, an approximate drop-out analysis was conducted by comparing the characteristics of the respondents in the current study with statistics from the Swedish Public Employment Service. That analysis showed that respondents with a high educational level were overrepresented in the current study [14].

Individuals who seldom spent time with close family, such as parents, sisters or brothers, or adult children, had higher odds of mental ill-health (OR = 1.16, 95% CI, 0.81–1.67), as shown in Table 3. Further, compared to individuals who frequently spent time with friends, individuals who rarely met with friends had higher odds of experiencing mental ill-health (OR = 1.70, 95% CI, 1.03–2.82).

Individuals who did not frequently attend social/community meetings or activities in an organisation or group such as a sports association or another kind of association, a church, a mosque, or women's or men's meetings, had higher odds for mental ill-health (OR = 1.58, 1.1–

**Table 1. Frequencies and percentages for the background variables.**

| Variables | | Frequences | Percentages |
|---|---|---|---|
| **Age** | | 336 | 50.5% |
| | 18–34 years | 176 | 26.5% |
| | 35–44 years | 107 | 16.1% |
| | 45–54 years | 43 | 6.5% |
| | 55–64 years | 3 | 0.5% |
| | 65–80 years | 336 | 50.5% |
| **Educational level** | | | |
| | Low educational level | 171 | 26.0% |
| | Medium educational level | 156 | 23.7% |
| | High educational level | 331 | 50.3% |
| **Gender** | | | |
| | Man | 461 | 69.3% |
| | Woman | 204 | 30.7% |
| **Crowded living** | | | |
| | No | 291 | 49.1% |
| | Yes | 302 | 50.9% |
| **Secure housing** | | | |
| | Yes | 192 | 30.7% |
| | No | 433 | 69.3% |

2.28) than individuals who did; see Table 3. All mentioned associations were adjusted for age, sex, educational level, crowded living, and housing conditions.

Individuals who seldom spent time with close family members, as well as seldom attending meetings or activities in an organisation or different groups, had higher odds of mental ill-health (OR = 2. 74, 95% CI, 1.29–5.83); see Table 4. Those answering "seldom" either with regard to being together with close family members, or with regard to meeting and spending time with friends or relatives, or attending meetings or activities in an organisation or different groups, had higher odds of mental ill-health (OR = 1.89, 95% CI, 1.23–2.92); see Table 4.

## Discussion

The results showed an association between seldom spending time with friends, as well as seldom attending meetings and activities, and suffering from mental ill-health. No association

**Table 2. Frequencies and percentages for the outcome and exposure variables.**

| | Yes n [%] | No n [%] |
|---|---|---|
| Outcome | | |
| Risk for mental illness [GHQ-12 ≥3] | 285 (0.47) | 319 (0.53) |
| Exposure | | |
| Seldom spend time with close family, such as parents, sisters or brothers, or children over 18 years old. | 253 (41.3) | 360 (58.7) |
| Seldom meet and spend time with friends or relatives. | 100 (15.9) | 530 (84.1) |
| Seldom spend spare time attending meetings or activities in an organisation or group, such as a sports association or another kind of association, a church, a mosque, or women's or men's meetings. | 313 (50.0) | 313 (50.0) |

**Table 3. The association between different aspects of social relations and mental ill-health.**

| Independent variable | OR 95%(CI)[a] | OR 95% (CI)[b] |
|---|---|---|
| Seldom spend time with close family, such as parents, sisters or brothers, or children over 18 years old. | 1.19 [0,85–1,65] p-value 0.31 | 1.16 [0.81–1.67] p-value 0.42 |
| Seldom meet and spend time with friends or relatives. | 1.72 (1.10–2.68) p-value 0.02 | 1.70 (1.03–2.82) p-value 0.04 |
| Seldom spend spare time attending meetings or activities in an organisation or group, such as a sports association or another kind of association, a church, a mosque, or women's or men's meetings. | 1.41 (1.02–1.94) p-value 0.04 | 1.58 (1.10–2.28) p-value 0.01 |

[a] Crude.

[b] Adjusted for age, sex, educational level, crowded living, and housing conditions.

was found between seldom being together with close family members and mental ill-health, however. Furthermore, the results suggest that seldom having social relations of the kind studied increases the risk of mental ill-health to a high degree as compared to frequently experiencing social relations regardless of frequency or type of social relations. However, there is a need to mention that this is seen within a group where a high proportion of the NAR have a higher education level. We could only assume that the association would be similar or stronger if the educational level was more equal.

A number of earlier studies have also found an association between social relations and mental health among refugees [1–3]. We know from earlier research that refugees frequently suffer from mental ill-health after arriving in the host countries [15] and that the mental health of refugees may by influenced by the migration experience, as a result of "pre-migration trauma and post-migration living difficulties" [16]. Some studies indicate a gradual improvement in symptoms over a period of a decade, to the point where the observed prevalence rates of mental disorder become lower than those of the general population of the host country [17,18], while other studies have found prevalence rates higher than those of the general population [19,20].

At the same time, we know from earlier research that refugees are challenged socially after arrival in the host countries and could suffer from social exclusion [6], facing language barriers that inhibit social relations [5] as well as another kind of social life that differs from the one in the home country [7]. We are also aware that different kinds of meeting places, such as churches and other places for religious activities [1,5], could help the NAR to establish social inclusion and thereby also integrate better in the host country. We are, furthermore, highly

**Table 4. The association between overall social relations and mental ill-health.**

| Social relations overall | OR* 95%(CI)[a] | OR* 95%(CI)[b] |
|---|---|---|
| Answering "seldom" to all three social relation questions. | 2.29 (1.16–4.51) p-value 0.02 | 2.74 (1.29–5.83) p-value 0.01 |
| Answering "seldom" to two of the three social relation questions. | 1.65 (1.03–2.62) p-value 0.04 | 1.58 (0.95–2.65) p-value 0.08 |
| Answering "seldom" to one of the three social relation questions. | 1.69 (1.14–2.50) p-value 0.01 | 1.89 (1.23–2.92) p-value 0.00 |

a Crude

b Adjusted for age, gender, educational level, crowded living, and housing conditions.

aware of the importance of good mental health for the newly arrived with regard to integration into the host country [21]. This is particularly important today since the refugees that have recently arrived have a lower educational level compared to those arriving earlier and therefore experience greater challenges in regard to accessing the Swedish labour market [16]. Hence, social relations could be seen as important for improving mental health among the NAR in Sweden and consequently also the integration into the society. The relevant stakeholders in Sweden should therefore endeavour to find meeting points for NAR that could enhance social relations and help build deeper contacts for the future.

The data collection took place closely in time after the permissions to stay in Sweden were granted and this could be seen as a strength of the study. The response rate for the study could be assessed as low when compared with the response rate of the last regional public health survey in Sweden [22], and this could be seen as a limitation of the current study. But when the response rate among those with a country of birth outside of Europe is compared between the regional public health survey and the current study, the rate of the latter is slightly higher. Also, when considering the current study in relation to other studies with similar populations, the current response rate is either in line with those studies or even higher [22,23]. Unfortunately, it was not possible to conduct any proper drop-out analysis since all respondents were anonymous. However, an approximate drop-out analysis was conducted by comparing the characteristics of the respondents in the current study with statistics from the Swedish Public Employment Service. That analysis showed that respondents with a high educational level were overrepresented in the current study [14]. This has to be considered a limitation, although we assume that the association between different aspects of social relations and mental ill-health would be similar or stronger if more NAR from lower educational levels were included. This could be assumed since we know from earlier studies that health and well-being are linked to educational level [24].

Moreover, the choice of confounders for the current study needs to be addressed. Crowded living was chosen on the basis of having shown itself to be linked to mental ill-health among NAR [8], but its association with social integration could be discussed. However, recent research tells of the link between integration and having a flat of your own [7]. Therefore, it could be argued that crowded living has an association with both the outcome and the exposure in the current study. Adjusting for the country of origin could also have been beneficial, but, unfortunately, we do not have this information from the survey.

Another limitation might be the way "social relations" overall were derived. This is important to know, as the categories including the answer "seldom" to the questions about social relations were grouped as one, two, or three "seldom" answers. We have thus not conducted separate analyses for type of social relation in the derived variable but tried to study a trend by combining different answers. The result of this indicates that irrespective of the kind of social relations (close family, and/or friends and relatives, and/or community activities) that the participants engaged in "seldom", there is an association between seldom having social relations and mental ill-health. There is, furthermore, an association between answering "seldom" to all questions concerning social relations and an increased risk of mental ill-health. The results indicate a clear difference between those being socially active and those being socially inactive. Furthermore, the adjusted analyses included age divided into age categories in order to determine the effect of different categories on the outcome. We have, however, also adjusted for age as a quantitative variable but this did not change the results notably.

The GHQ-12 has been shown to be a good tool for assessing mental health since it has been validated [12,25] and found to be a secure measure of mental health [12], and the validity is not influenced by gender, age, or educational level [12]. It has been adopted by the World Health Organization (WHO) in a study of psychological disorders in primary healthcare and

has been deemed the best validated among similar screening tools [12,13,26]. To the best of our knowledge, the GHQ-12 is not especially adapted for studies that focus on refugees; it has, however, been used in other studies concerning refugees before [27,28]. In line with Goldberg's original recommendation, a score sum of $\geq 3$ was used in the current study. It could be discussed whether this scoring is relevant for the studied refugee population, but the same scoring has been used in another study which also focuses on refugees [28].

## Conclusions

The study suggests a link between spending time with friends, as well as finding meeting spots, and mental health, for newly arrived refugees in the southernmost county of Sweden. Since this study was cross-sectional and had a high proportion of NAR with a high educational level, we suggest that further studies with a longitudinal design are needed to confirm this finding.

## Author Contributions

**Conceptualization:** Elisabeth Mangrio, Slobodan Zdravkovic.

**Data curation:** Mathias Grahn.

**Formal analysis:** Mathias Grahn, Slobodan Zdravkovic.

**Funding acquisition:** Slobodan Zdravkovic, Carin Cuadra.

**Investigation:** Elisabeth Mangrio.

**Methodology:** Elisabeth Mangrio, Mathias Grahn, Slobodan Zdravkovic.

**Project administration:** Elisabeth Mangrio.

**Validation:** Elisabeth Mangrio, Mathias Grahn.

**Writing – original draft:** Elisabeth Mangrio.

**Writing – review & editing:** Elisabeth Mangrio, Mathias Grahn, Slobodan Zdravkovic, Carin Cuadra.

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
