## [Decision Letter · Decision Letter 0]

21 Jan 2022

PGPH-D-21-00113

Social relations and mental ill-health among newly arrived refugees in Sweden: a cross-sectional study

Dear Dr. Mangrio,

Thank you for submitting your manuscript to PLOS Global Public Health. After careful consideration, we feel that it has merit but does not fully meet PLOS Global Public Health’s publication criteria as it currently stands. Therefore, we invite you to submit a revised version of the manuscript that addresses the points raised during the review process.

We look forward to receiving your revised manuscript.

Kind regards,

Joseph El-Khoury, MD MSc FRCPsych

Academic Editor

Journal Requirements:

1. Please include additional information regarding the survey or questionnaire used in the study and ensure that you have provided sufficient details that others could replicate the analyses. For instance, if you developed a questionnaire as part of this study and it is not under a copyright more restrictive than CC-BY, please include a copy, in both the original language and English, as Supporting Information.

2. In the online submission form, you indicated that "Data can be requested from the authors.". All PLOS journals now require all data underlying the findings described in their manuscript to be freely available to other researchers, either 1. In a public repository, 2. Within the manuscript itself, or 3. Uploaded as supplementary information.

3. Please amend your detailed Financial Disclosure statement. This is published with the article, therefore should be completed in full sentences and contain the exact wording you wish to be published.

ii). State the initials, alongside each funding source, of each author to receive each grant.

iii). State what role the funders took in the study. If the funders had no role in your study, please state: “The funders had no role in study design, data collection and analysis, decision to publish, or preparation of the manuscript.”

iv). If any authors received a salary from any of your funders, please state which authors and which funders.

Additional Editor Comments (if provided):

Dear Authors

I would like first to apologise for the delay in proving you with an initial decision on your manuscript. This is due to the difficulty in securing reviewers. On this basis I have decided to request you to revise your paper based on the fairly comprehensive comments on one reviewer.

Reviewers' comments:

Reviewer's Responses to Questions

**Comments to the Author**

1. Does this manuscript meet PLOS Global Public Health’s publication criteria? Is the manuscript technically sound, and do the data support the conclusions? The manuscript must describe methodologically and ethically rigorous research with conclusions that are appropriately drawn based on the data presented.

Reviewer #1: Yes

2. Has the statistical analysis been performed appropriately and rigorously?

Reviewer #1: Yes

3. Have the authors made all data underlying the findings in their manuscript fully available (please refer to the Data Availability Statement at the start of the manuscript PDF file)?

Reviewer #1: No

4. Is the manuscript presented in an intelligible fashion and written in standard English?

Reviewer #1: No

5. Review Comments to the Author

Reviewer #1: The authors describe a study among newly arrived refugees (NARs) in Sweden. NARs who had more frequent contact with friends and family, as well as NARs who were more involved in social groups, were at lower risk of experiencing mental ill-health (on the GHQ-12), after adjusting for other factors. This article addresses an important area of study, as research continues to show strong associations between social support and mental health and wellbeing. The manuscript could benefit from a stronger justification in the Introduction, including more context on the state of the extant literature and gaps in the research. I also suggest including additional recommendations for future research and intervention in the Discussion.

There are several grammatical errors throughout the manuscript, as well as sentences that could benefit from restructuring - too many to type out line by line in this review. During revision, the authors would benefit from having a native English speaker provide a thorough review/proofing of the written English. I describe specific comments by section below.

Abstract

- In the background subsection, I would replace the second sentence with a sentence indicating the gap in the literature. As it is written, the second and third sentences of this subsection are repetitive of one another.

- In the methods subsection, it is unclear what is meant by "the civic and health information"

- In the results subsection, I would make sure to use a consistent number of decimal points when presenting statistics. Additionally, it is necessary to include the level of confidence for the CI -- 95% CIs. I also find the way the data are presented confusing. For example, the first sentence could be rephrased as follows: "Compared to individuals who frequently spent time with friends, individuals who rarely met with friends had higher odds of experiencing mental ill-health (OR=1.70, 95% CI, 1.03-2.82). I would rephrase the second sentence similarly.

Introduction:

- There are several grammatical and some spelling errors throughout the Introduction that reduce the clarity of concepts/ flow of ideas.

- The authors mention religion in the first paragraph -- involvement in religious life can protect mental health in two distinct ways, through social relations/community and through the actual spiritual-religious belief systems/practice. If religion is included in this study, the authors should clarify that they are only focused on the social/community aspect, or they should expand the focus of the study to include religious involvement alongside social involvement. Why do the authors singularly focus on religion without including other types of social groups?

- I recommend including references in the last paragraph. Additionally, the Introduction is lacking in a discussion of the gap in the existing literature in this area. What specifically does this study add to other research on social relationships and mental health among newly arrived refugees? What is new about this study?

- I would briefly note where Scania is in Sweden (e.g., "among NAR in Scania, the southernmost county in Sweden.")

Material and Methods:

- As noted in the Abstract comments, it is unclear what the authors mean by "participated in the civic and health information" -- I do not understand how someone participates in "information".

- I would include additional detail on the inclusion/exclusion criteria; where were most refugees in the public integration support program from? Was the present study part of a larger study? (I ask because the authors mention several other questionnaires -- sleep, general health). I would either briefly explain the larger study, or I would limit the description of measures to the ones that are actually relevant to the present study.

- Were the questionnaires translated into Dari and Arabic? What did this translation process look like? Were these paper-based or computer-based questionnaires?

- I recommend citing psychometric properties of the GHQ-12; has it been validated for refugees from these countries? This is briefly mentioned in the Discussion.

- Were the social relations questions adapted from existing measures? I found the grouping and categorization of these questions confusing.

- There is a clear focus on religion in the Introduction section, but then the Methods refers to several other types of group involvement. I would expand on these in the Introduction, since it currently reads as if religion is a primary focus.

- Why was age categorized rather than included as a continuous variable? I recommend leaving as continuous if possible.

- It might be worth justifying why crowded living and housing conditions were included in the analyses -- perhaps comment in the Introduction on how these variables might impact associations between social integration and mental health. (Particularly since crowded living would naturally increase interactions with family).

- It's generally best not to use odds ratios when the outcome variable is not rare (Davies, Crombie & Tavakoli, 1998). Did the authors consider fitting a different model (e.g., Poisson regression instead of logistic)?

- Did the authors consider also adjusting for host country? I would expect culture to potentially play a role in observed associations between social integration and mental health.

- Did the questionnaire ask about gender or sex (authors refer to gender, but only include male and female categories)? I would be careful with this distinction.

Results:

- I would re-title the rows of Table 1 for parallel structure and to make the titles more concise. I would also include the total N and percentages in this table. I also recommend including other variables in the descriptive statistics table (mean/SD age, education level, crowded living, housing conditions).

- It might be worth including the results of both unadjusted and adjusted models in Table 2 / the results section.

- Same comments on presentation of written results as described in the Abstract comments. As the data are presented, it is difficult to interpret the results. This section would benefit from grammatical/structuring revisions for clarity.

- If you use the term "significant", I would note whether you are referring to statistical significance. (Be cautious of this throughout the manuscript)

Discussion:

- Early on in the Discussion, the authors note "no significant association" for time with family and mental health - however, the Conclusions section states that there was an association between these variables.

- The authors note an approximate drop-out analysis in the limitations section - I recommend that any analyses such as this be introduced first in the Results section.

- The authors address some of my earlier questions about GHQ-12 validity and cultural adaptation in the limitations section. I would also mention some of this validity research in the Methods.

6. PLOS authors have the option to publish the peer review history of their article (what does this mean?). If published, this will include your full peer review and any attached files.

**Do you want your identity to be public for this peer review?** For information about this choice, including consent withdrawal, please see our Privacy Policy.

Reviewer #1: No

---

## [Editor Report · Decision Letter 1]

7 Jun 2022

Social relations and mental ill-health among newly arrived refugees in Sweden: a cross-sectional study

PGPH-D-21-00113R1

Dear Dr. Mangrio,

We are pleased to inform you that your manuscript 'Social relations and mental ill-health among newly arrived refugees in Sweden: a cross-sectional study' has been provisionally accepted for publication in PLOS Global Public Health.

Best regards,

Joseph El-Khoury, MD MSc FRCPsych

Academic Editor